# Screening and biodiversity analysis of cultivable inorganic phosphate–solubilizing bacteria in the rhizosphere of *Hydrilla verticillata*

**Yong Li**[1]*, **Huan Liu**[1], **Xintao Yu**[1], **Sidan Gong**[1], **Zhilian Gong**[2]

**1** Faculty of Geosciences and Environmental Engineering, Southwest Jiaotong University, Chengdu, China,
**2** School of Food and Biological Engineering, Xihua University, Chengdu, China

\* liyong@swjtu.edu.cn

☗ OPEN ACCESS

**Data Availability Statement:** All relevant data are available from the National Center for Biotechnology Information at https://www.ncbi.

## Abstract

The inorganic phosphate–solubilizing bacteria (IPB) in the rhizosphere of *Hydrilla verticillata* can convert insoluble inorganic phosphorus in the environment into soluble phosphorus that can be directly absorbed and utilized by *Hydrilla verticillata*. In this research, the roots and rhizosphere sediments of *Hydrilla verticillata* were collected from high–organic matter urban landscape water. The National Botanical Research Institute's Phosphate growth medium (NBRIP medium) was used to screen for efficient cultivable IPB. The 16S rRNA gene sequence analysis was used to determine the taxonomic affiliation of the strains, and ammonium molybdate spectrophotometry was used to detect the phosphate-solubilizing ability of the strains. The results show that a total of 28 IPB strains with good phosphate-solubilizing effect are obtained from the roots and rhizosphere sediments of *Hydrilla verticillata*. These IPB strains belong to two phyla, four orders, seven classes, nine families, and nine genera. Among these, *Bacillus* and *Acinetobacter* are the dominant genera, and the strains SWIH-7, SWIP-6, SWIP-7, SWIP-13, SWIP-15 and SWIP-16 are potential new species. The IPB strains isolated and screened in this research are rich in diversity, with potential new species and stable phosphate-solubilizing characteristic. These IPB strains are suitable for further development as microbial bacterial agents, which can be applied to promote the recovery of submerged plants in polluted water with high–organic matter, treatment of polluted water and ecological restoration of water.

## Introduction

The problem of water pollution has become one of the most important environmental problems in China and the world, which seriously threatens the safety of the water environment on which human beings depend. High–load sediment pollution is one of the water pollution, and controlling sediment endogenous pollution is the key concern of water environmental protection and management [1]. Polluted water remediation methods include physical, chemical, biological and combined methods. Compared with physical and chemical methods, biological

nlm.nih.gov/. Accession numbers of IPB strains are within the manuscript.

**Funding:** This research was funded by the Key Laboratory Open Project of Southwest Jiaotong University (ZD2022210017). This research was funded by Student Research Training Program (2023145).

**Competing interests:** The authors have declared that no competing interests exist.

methods have the advantages of low environmental disturbance, no secondary pollution, and are conducive to the self-sustaining balance of aquatic ecosystems, and have broad application prospects in the treatment of polluted water [2].

*Hydrilla verticillata* is a perennial submerged plant that grows and distributes in lakes, ponds and rivers. *Hydrilla verticillata* is adapted to the environment faster after artificial transplantation and grow faster compared with other submerged plants, so the absorption of nutrient salts, such as nitrogen and phosphorus, is also more desirable [3], which can be used for the restoration of water. However, *Hydrilla verticillata* is highly resistant and invasive [4]. In high–organic matter water, the growth of *Hydrilla verticillata* is inhibited. Sediment is one of the main factors affecting the growth of submerged plants and plays a decisive role in the growth, morphology and distribution characteristics of submerged plants [5, 6]. Submerged plants, such as *Hydrilla verticillata*, are often plagued by problems such as high–load sediment stress during restoration [7]. The high nutrient salts in sediment can stress the seed germination and seedling growth of submerged plants, and even have toxic effects, thus affecting the ecological recovery process of the whole water. The high–organic matter content will cause the sediment to become reductive silt, which has a stressful effect on submerged plants and is not conducive to the survival and germination of submerged plants [8]. To overcome this drawback, there is an urgent need to develop a technology that can promote the conversion of insoluble phosphorus in the water into soluble phosphorus that can be absorbed and utilized by *Hydrilla verticillata*, as well as promote the growth of *Hydrilla verticillata* in high–organic matter polluted water. Inorganic phosphate–solubilizing bacteria (IPB) in the rhizosphere of *Hydrilla verticillata* can convert insoluble inorganic phosphorus in the environment into soluble phosphorus that can be easily absorbed and utilized [9]. *Hydrilla verticillata* absorbs phosphorus through roots and leaves to promote their own growth.

At present, more studies on IPB in the environment have been reported, and these studies mainly focused on the research direction of IPB in soil crop rhizosphere [10]. Researchers have isolated bacterial strains with good phosphate-solubilizing ability from rhizosphere soil of crops and have conducted studies on phosphate-solubilizing characteristic and proliferation characteristic of the strains [11–15]. Although more researchers have studied IPB in soil and other environments, relatively few studies have been conducted on IPB in the rhizosphere of submerged plants. Studies on IPB in more polluted water, such as landscape water, lakes, and black and odorous water, are scarce. In recent years, the effect of submerged plants on the phosphorus content in their rhizosphere soil sediments has attracted widespread attention. Wang Lizhi et al. [16] found that the alternating growth process of *Hydrilla verticillata* in the decaying phase and miner's grass in the growing phase caused the migration of phosphorus from the aqueous phase to the sediment phase, thus keeping the phosphorus concentration of each form in the aqueous phase at a relatively low level. However, the effect of IPB on the phosphorus content in the rhizosphere sediment of submerged plants and the promotion effect of IPB on the growth and development of submerged plants are less studied.

Therefore, this research aimed to use the NBRIP medium to screen for efficient cultivable IPB from the roots and rhizosphere sediments of *Hydrilla verticillata*, in high–organic matter urban landscape water in Chuangzhi Park, Pidu District, Chengdu, Sichuan Province, explore their biodiversity, and clarify the phosphate-solubilizing capacity of IPB in the rhizosphere of *Hydrilla verticillata*. The IPB screened in this research can be developed into microbial bacterial agents to act on the rhizosphere of submerged plants, providing phosphorus and other nutrients to submerged plants, which is important for the restoration of submerged plants in polluted water with high–organic matter, the treatment of polluted water and the ecological restoration of water.

## Materials and methods

### Source of experimental samples

Cultivable IPB were isolated and screened from the roots and rhizosphere sediments of *Hydrilla verticillata* collected from an urban landscape water in Chuangzhi Park, Pidu District, Chengdu, Sichuan Province (E103˚54′, N30˚48′).

### Culture medium

The inorganic phosphorus medium (NBRIP) [17]: glucose, 10 g; $Ca_3(PO_4)_2$, 5 g; $MgCl_2 \cdot 6H_2O$, 5 g; $MgSO_4 \cdot 7H_2O$, 0.25 g; KCl, 0.2 g; $(NH_4)_2SO_4$, 0.1 g; distilled water, 1 L; with a pH 7.0–7.5. $Ca_3(PO_4)_2$ was required to be sterilized separately; NBRIP solid medium was added with 20 g agar.

The Luria-Bertani (LB) medium: NaCl, 10 g; yeast paste, 5 g; peptone, 10 g; agar, 20 g; and distilled water, 1 L; with a pH 7.0–7.2.

The phosphorus-deficient medium [18]: $CH_3COONa \cdot 3H_2O$, 3.23 g; $Na_2HPO_4 \cdot 2H_2O$, 23 mg; $NH_4Cl$, 152.8 mg; $MgSO_4 \cdot 7H_2O$, 81.12 mg; $K_2SO_4$, 17.83 mg; $CaCl_2 \cdot 2H_2O$, 11 mg; 4-(2-hydroxyethyl)-1-piperazineethanesulfonic acid (HEPES), 7 g; trace element solution, 2 mL; and distilled water, 1000 mL; with a pH = 7.

### Isolation and screening of IPB

The roots with good growth conditions and mud were selected from the collected *Hydrilla verticillata* and cut up with scissors. Firstly, 10 g cut mud-bearing roots were added to a conical flask containing 100 mL of sterile water and glass beads. The conical flask was placed in a shaking incubator at 600 rpm and 28˚C for two hours to form a sediment suspension. Then, 1 mL of the suspension was taken and 9 mL of sterile distilled water was added to it to form a $10^{-1}$ bacterial suspension. The suspension was taken and diluted step by step to $10^{-2}$–$10^{-9}$. Then, 0.1 mL of each was taken and spread on the NBRIP solid plate medium, with three replicates of each concentration. Next, colony with obvious phosphate-solubilizing ring on the NBRIP solid medium was picked for lineation and separation by repeated lineation for more than four times. After the single colony was confirmed as pure culture by morphological characteristic and microscopic examination, the LB slant medium was inoculated, cultured and stored in a 4˚C refrigerator for further use.

### Detection of phosphate-solubilizing characteristic of IPB

The strains were inoculated in a phosphorus-deficient medium and incubated at 28˚C for 24 h. The culture solution was poured into a centrifuge tube and centrifuged at 4000 rpm for five minutes. After centrifugation, the supernatant was discarded, and distilled water was added and mixed with a vortex to prepare a bacterial suspension with an $OD_{600}$ (Optical Density) value of about 0.2. Then, 2 mL of the suspension was added to 100 mL of freshly sterilized NBRIP liquid medium and incubated at 28˚C for seven days. Each treatment was replicated three times. The uninoculated medium was used as the control. The concentration of soluble reactive phosphate (SRP) in the culture solution from three to seven days was determined by the molybdenum antimony anti-colorimetric method [19], the pH of the culture solution was measured using a pHS-3C acidity meter, the concentration of dissolved inorganic phosphorus (DIP) was calculated from the standard curve, with three replicates per group, phosphate-solubilizing efficiency was calculated by the ratio of the concentration of dissolved inorganic phosphorus in the culture solution to the original phosphorus concentration, and the

concentration of dissolved inorganic phosphorus of the corresponding blank control was removed from all data.

## Molecular biology identification of strains

The total bacterial DNA was extracted with the bacterial genomic DNA extraction kit from Biotech Bioengineering (Shanghai, China.) Co., Ltd, following the kit instructions, and polymerase chain reaction (PCR) amplification of 16S rRNA was performed with universal primers 27F (5′–AGAGTTTGATCCTGGCTCAG–3′) and 1492R (5′–GGTTACCTTGTTACGACTT–3′) [20]. The PCR reaction system (20 μL) was as follows: 10× Ex Taq buffer, 2.0 μL; 5U Ex Taq, 0.2 μL; 2.5mM dNTP mix, 1.6 μL; 27F, 1 μL; 1492R, 1 μL; DNA, 0.5 μL; and ddH$_2$O, 13.7 μL. The PCR reaction conditions were as follows: denaturation at 95˚C for 30 s, annealing at 56˚C for 30 s, extension at 72˚C for 90 s, 25 cycles, pre-denaturation before reaction for 5 min (95˚C), and extension at the end of the cycle for 10 min (72˚C). The PCR amplification products were detected by 0.8% agarose gel electrophoresis. The PCR products were purified using the kit and then sequenced using a 3730XL sequencer [Biotech Bioengineering (Shanghai, China.) Co., Ltd.]. The obtained 16S rRNA full sequences were submitted to the Basic Local Alignment Search Tool (BLAST) in the National Center for Biotechnology Information (NCBI; https://www.ncbi.nlm.nih.gov/) to identify the species classification of the strains. The neighbor-joining method was used to construct a phylogenetic tree [21].

## Statistical analysis

Statistical analysis was performed using SPSS statistical software (version 20.0, IBM, Armonk, NY, USA). The one-way analysis of variance (ANOVA) was used to analyze the difference of DIP among different IPB strains. Significance levels were using $P = 0.05$ in all statistical analyses. Duncan method was used for POST-HOC test.

## Results and discussion

### Separation screening results and biodiversity analysis

A total of 28 strains were obtained from the roots and rhizosphere sediments of *Hydrilla verticillata* with a good phosphate-solubilizing effect.

These 28 strains were sequenced and the sequencing results were submitted to the NCBI database for the BLAST homology analysis, and the results are depicted in Table 1. The phylogenetic evolutionary tree is shown in Fig 1.

The results show that the isolated strains belong to two phyla: *Proteobacteria* and *Firmicutes*, involving four orders (*Betaproteobacteria*, *Alphaproteobacteria*, *Gammaproteobacteria* and *Bacilli*), seven classes (*Burkholderiales*, *Enterobacteriales*, *Caulobacterales*, *Moraxellales*, *Hyphomicrobiales*, *Aeromonadales* and *Bacillales*), nine families (*Comamonadaceae*, *Enterobacteriaceae*, *Moraxellaceae*, *Aeromonadaceae*, *Caulobacteraceae*, *Yersiniaceae*, *Bacillales Family XII. Incertae Sedis*, *Rhizobiaceae* and *Bacillaceae*), and nine genera (*Bacillus*, *Asticcacaulis*, *Acinetobacter*, *Aeromonas*, *Serratia*, *Exiguobacterium*, *Comamonas*, *Ciceribacter* and *Priestia*). The genus *Bacillus* has the highest number of IPB, with nine strains, followed by the genus *Acinetobacter*, with five strains. Among these, *Bacillus* and *Acinetobacter* are the dominant genera, accounting for 32.1% and 17.9%, respectively.

A research revealed [22] that strains belonging to the genera *Bacillus*, *Acinetobacter* and *Serratia* are the more common strains of phosphate-solubilizing bacteria with phosphate-solubilizing functions. This research isolates *Asticcacaulis*, *Aeromonas*, *Exiguobacterium*, *Comamonas*, *Ciceribacter* and *Priestia* as IPB, which are rarely reported.

**Table 1. Online BLAST comparison results of 16S rRNA gene sequences of IPB.**

| Strain Number | Similar Strain [a] | Gene Identity (%) [b] | Identification Results | Registry Number [c] |
|---|---|---|---|---|
| SWIP-1 | *Priestia megaterium* (NBRC:15308;ATCC:14581) | 99.72 | *Priestia megaterium* | ON479625 |
| SWIP-3 | *Bacillus thuringiensis* (IAM:12077;ATCC:10792) | 99.72 | *Bacillus thuringiensis* | ON479626 |
| SWIP-5 | *Asticcacaulis excentricus* (DSM:4724;ATCC:15261) | 99.41 | *Asticcacaulis excentricus* | ON479627 |
| SWIP-6 | *Acinetobacter proteolyticus* (CCM:8640;CCUG:67965) | 98.29 | *Acinetobacter proteolyticus* | ON479637 |
| SWIP-7 | *Ciceribacter naphthalenivorans* (NBRC:107585;KCTC:23252) | 98.13 | *Ciceribacter naphthalenivorans* | ON571639 |
| SWIP-8 | *Priestia megaterium* (NBRC:15308;ATCC:14581) | 99.79 | *Priestia megaterium* | ON479638 |
| SWIP-9 | *Serratia surfactantfaciens* (KCTC:42987;CCTCC:AB2015384) | 99.65 | *Serratia surfactantfaciens* | ON571640 |
| SWIP-10 | *Aeromonas media* (ATCC:33907;BCRC:14129) | 99.86 | *Aeromonas media* | ON571641 |
| SWIP-11 | *Bacillus cereus* (ATCC:14579;BCRC:10603) | 99.23 | *Bacillus cereus* | ON571642 |
| SWIP-12 | *Aeromonas media* (ATCC:33907;BCRC:14129) | 99.86 | *Aeromonas media* | ON479639 |
| SWIP-13 | *Acinetobacter haemolyticus* (ATCC:17906;CCUG:888) | 98.30 | *Acinetobacter haemolyticus* | ON479640 |
| SWIP-15 | *Acinetobacter haemolyticus* (ATCC:17906;CCUG:888) | 98.30 | *Acinetobacter haemolyticus* | ON479641 |
| SWIP-16 | *Acinetobacter proteolyticus* (CCM:8640;CCUG:67965) | 98.17 | *Acinetobacter proteolyticus* | ON479642 |
| SWIP-18 | *Aeromonas media* (ATCC:33907;BCRC:14129) | 99.51 | *Aeromonas media* | ON479628 |
| SWIV-1 | *Bacillus velezensis* (CCUG:50740;BCRC:17467) | 99.86 | *Bacillus velezensis* | ON479633 |
| SWIV-2 | *Bacillus thuringiensis* (IAM:12077;ATCC:10792) | 99.52 | *Bacillus thuringiensis* | ON479629 |
| SWIV-3 | *Bacillus thuringiensis* (IAM:12077;ATCC:10792) | 99.93 | *Bacillus thuringiensis* | ON479643 |
| SWIV-7 | *Exiguobacterium profundum* (CCUG:50949;DSM:17289) | 98.96 | *Exiguobacterium profundum* | ON571643 |
| SWIV-10 | *Bacillus toyonensis* (CECT:876;NCIMB:14858) | 99.79 | *Bacillus toyonensis* | ON479644 |
| SWIV-11 | *Exiguobacterium profundum* (CCUG:50949;DSM:17289) | 98.68 | *Exiguobacterium profundum* | ON571644 |
| SWIV-12 | *Bacillus toyonensis* (CECT:876;NCIMB:14858) | 99.50 | *Bacillus toyonensis* | ON479645 |
| SWIH-1 | *Priestia aryabhattai* B8W22 (MTCC:7755;JCM:13839) | 99.28 | *Priestia aryabhattai* | ON479646 |
| SWIH-2 | *Priestia aryabhattai* B8W22 (MTCC:7755;JCM:13839) | 99.78 | *Priestia aryabhattai* | ON479647 |
| SWIH-3 | *Comamonas aquatica subsp. rana* (KCTC:12879;DSM:18981) | 100 | *Comamonas aquatica subsp. rana* | ON479630 |
| SWIH-5 | *Aeromonas media* (ATCC:33907;BCRC:14129) | 99.72 | *Aeromonas media* | ON479631 |
| SWIH-7 | *Acinetobacter haemolyticus* (ATCC:17906;CCUG:888) | 97.84 | *Acinetobacter haemolyticus* | ON479632 |
| SWIH-8 | *Comamonas aquatica subsp. rana* (KCTC:12879;DSM:18981) | 100 | *Comamonas aquatica subsp. rana* | ON479648 |

(*Continued*)

**Table 1.** (Continued)

| Strain Number | Similar Strain [a] | Gene Identity (%) [b] | Identification Results | Registry Number [c] |
|---|---|---|---|---|
| SWIH-9 | *Priestia aryabhattai B8W22* (MTCC:7755;JCM:13839) | 99.72 | *Priestia aryabhattai* | ON479649 |

Note:

[a]. Sequence with highest percentage of identity observed in EzBioCloud.

[b]. Percentage of identity with EzBioCloud analysis.

[c]. The accession number of IPB in NCBI.

According to the method described in a previous research [23], among 28 IPB strains screened, strains with greater than 98.65% 16S rRNA gene sequence similarity are classified as belonging to the same species. Therefore, strains SWIH-7, SWIP-6, SWIP-7, SWIP-13, SWIP-15 and SWIP-16 are tentatively identified as potential new species.

All 16S rRNA sequences of representative strains are selected from Table 1 for BLAST analysis on the NCBI website to search for sequences with high homology and construct phylogenetic tree. According to Fig 1, the evolutionary tree is divided into two large branches, and

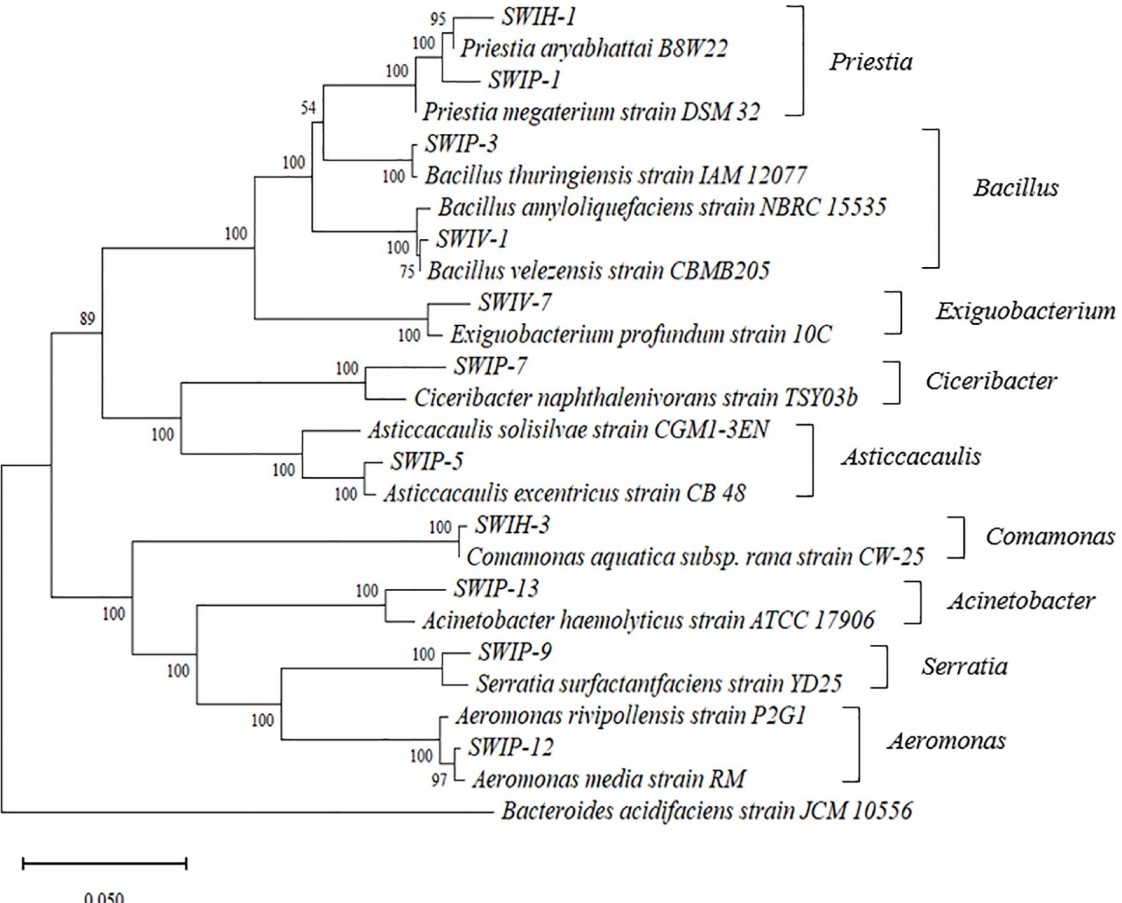

**Fig 1. Phylogenetic evolutionary tree constructed based on 16S rRNA gene sequences.** The numbers at the nodes indicate the bootstrap values based on the neighbor-joining analyses of 1000 resampled data sets. The scale bar indicates evolutionary distance.

each large branch is divided into two small branches. The first subclade of the first major contains three genera (*Priestia*, *Bacillus* and *Exiguobacterium*), and the second subclade contains two genera (*Ciceribacter* and *Asticcacaulis*). The first subclade of the second subclade includes one genera (*Comamonas*), and the second subclade includes three genera (*Acinetobacter*, *Serratia* and *Aeromonas*). The evolutionary distance and kinship of 28 IPB strains in the evolutionary tree are shown in Fig 1 and Table 1.

## Analysis of phosphate-solubilizing characteristic

After the screening and purification of the NBRIP medium, 28 IPB strains with a good phosphate-solubilizing effect were obtained. The $OD_{600}$, DIP and phosphate-solubilizing efficiency of the culture solution are provided in Table 2.

**Table 2. Incubation time, $OD_{600}$, DIP and phosphate-solubilizing efficiency.**

| Strain Number | Incubation Time (d) [i] | pH [ii] | $OD_{600}$ [iii] | DIP (mg/L) [iv] | Phosphate-solubilizing Efficiency |
|---|---|---|---|---|---|
| CK | - | 7.0 | - | 0.23 ± 0.01 f | - |
| SWIP-1 | 4 | 5.9 | 1.049 | 29.28 ± 1.32 c | 37% |
| SWIP-3 | 3 | 5.3 | 2.105 | 38.53 ± 1.74 b | 49% |
| SWIP-5 | 6 | 6.0 | 0.695 | 32.50 ± 1.63 c | 45% |
| SWIP-6 | 4 | 6.5 | 1.729 | 19.84 ± 0.79 d | 25% |
| SWIP-7 | 4 | 4.9 | 0.405 | 49.26 ± 1.48 ab | 62% |
| SWIP-8 | 5 | 5.0 | 1.457 | 44.67 ± 0.88 b | 56% |
| SWIP-9 | 6 | 6.2 | 0.221 | 25.83 ± 0.26 cd | 33% |
| SWIP-10 | 3 | 5.7 | 2.208 | 32.69 ± 0.33 c | 41% |
| SWIP-11 | 5 | 6.4 | 1.422 | 22.27 ± 0.35 d | 29% |
| SWIP-12 | 3 | 5.2 | 2.122 | 39.30 ± 0.39 b | 50% |
| SWIP-13 | 4 | 6.5 | 1.760 | 21.92 ± 0.66 d | 28% |
| SWIP-15 | 5 | 6.7 | 1.491 | 2.19 ± 0.05 e | 3% |
| SWIP-16 | 5 | 5.4 | 1.464 | 33.91 ± 1.21 c | 43% |
| SWIP-18 | 3 | 5.1 | 2.175 | 35.54 ± 0.89 bc | 45% |
| SWIV-1 | 5 | 6.6 | 1.195 | 14.26 ± 0.14 de | 18% |
| SWIV-2 | 5 | 6.1 | 1.165 | 31.48 ± 0.47 c | 39% |
| SWIV-3 | 4 | 5.5 | 1.714 | 33.37 ± 0.50 c | 42% |
| SWIV-7 | 4 | 6.2 | 2.263 | 24.45 ± 0.38 cd | 32% |
| SWIV-10 | 5 | 6.3 | 1.231 | 24.37 ± 0.49 cd | 30% |
| SWIV-11 | 5 | 5.2 | 1.373 | 39.15 ± 0.96 b | 49% |
| SWIV-12 | 5 | 5.8 | 1.356 | 32.61 ± 0.49 c | 41% |
| SWIH-1 | 5 | 5.1 | 1.486 | 41.28 ± 0.84 b | 51% |
| SWIH-2 | 5 | 5.3 | 1.438 | 37.55 ± 0.58 b | 49% |
| SWIH-3 | 4 | 6.7 | 1.748 | 2.94 ± 0.06 e | 4% |
| SWIH-5 | 3 | 6.1 | 2.408 | 28.48 ± 0.58 c | 36% |
| SWIH-7 | 4 | 6.4 | 1.757 | 20.83 ± 0.31 d | 26% |
| SWIH-8 | 4 | 6.0 | 1.804 | 29.13 ± 0.87 c | 37% |
| SWIH-9 | 4 | 4.8 | 1.463 | 55.74 ± 0.56 a | 71% |

Note: CK: not inoculated. i. Incubation time: the time corresponding to the best phosphate-solubilizing effect of the strain in NBRIP medium. ii. pH: the pH corresponding to the best phosphate-solubilizing effect of the strain in the NBRIP liquid medium. iii. $OD_{600}$: the OD corresponding to the best phosphate-solubilizing effect of the strain in the NBRIP liquid medium. iv. the maximum DIP value of sample after subtracting the CK DIP value in the NBRIP liquid medium. In the same column, different letters a, b and c indicate significant differences ($P < 0.05$), the same letters indicate non-significant differences ($P > 0.05$), and data containing letters such as ab are not significantly different from those containing a and those containing b.

As can be seen from Table 2, the pH of the culture solution of the strains with high phosphate-solubilizing ability is lower, while the pH of the culture solution of the strains with low phosphate-solubilizing ability is higher. Fewer strains show special characteristics, indicating that these strains could secrete some acidic substances during the culture process, resulting in a decrease in pH and the dissolution of insoluble phosphorus. According to the relationship between the concentration of dissolved inorganic phosphorus and pH of 28 IPB strains, the phosphate-solubilizing ability of IPB is significantly negatively correlated with the pH of the solution (R = -0.925, N = 28, $P < 0.01$). The lower the pH value of the culture solution, the higher the concentration of dissolved inorganic phosphorus is generally observed.

The results show that all 28 strains have phosphate-solubilizing characteristic, among which 25 strains, such as strains SWIP-1, SWIP-3 and SWIP-5, have phosphate-solubilizing efficiency of more than 20%. Among these, the strain SWIH-9 has the highest phosphate-solubilizing efficiency of about 71% and DIP of about 55.74 mg/L, which is significantly higher than other bacteria ($P < 0.05$). Seven strains, such as SWIP-1, SWIP-10, SWIV-2, SWIV-3, SWIV-12, SWIH-5 and SWIH-8, have phosphate-solubilizing efficiency between 36% and 42%, and the DIP is about 28.48–33.37 mg/L. Seven strains, such as SWIP-6, SWIP-9, SWIP-11, SWIP-13, SWIV-7, SWIV-10 and SWIH-7, have phosphate-solubilizing efficiency between 25% and 33%, and the DIP is about 19.84–25.83 mg/L.

This research isolated 28 strains of IPB of two phyla and nine genera from *Hydrilla verticillata*. Chen et al. [24] obtained 25 strains of endophytic bacteria of two phyla and eight genera from *Nymphaea tetragona* in the Guan Hall Reservoir in Beijing. The two phyla were *Gammaproteobacteria* and *Bacilli*, and the eight genera were *Pseudomonas*, *Enterobacter*, *Aeromonas*, *Klebsiella*, *Pantoea*, *Bacillus*, *Paenibacillus* and *Lactococcus*. Compared with the results of Chen et al., the diversity of IPB isolated in this research is richer. Wang Xiaodan et al. [25] isolated two strains from the rhizosphere soil of *Pennisetum alopecuroides*, inoculated the strains in the NBRIP liquid medium, and measured the phosphate-solubilizing ability of strains WXD 2–1 and WXD 2–2 to be 26.99 mg/L and 31.99 mg/L, respectively. Compared with the results of Wang et al., the phosphate-solubilizing ability of the strain SWIH-9 obtained in this research is 55.74 mg/L, which indicates that the IPB strains obtained in this research have higher phosphate-solubilizing activity. Gong Ruihong et al. [26] isolated 42 strains of phosphate-solubilizing bacteria from the rhizosphere soil of four dominant plant species in the desert steppe of Siziwang Banner, and isolated phosphate-solubilizing bacteria using NBRIP medium with 10 g/L $Ca_3(PO_4)_2$. The concentration of dissolved inorganic phosphorus of the phosphate-solubilizing bacteria was 0.95–77.57 mg/L. The phosphate-solubilizing ability of IPB isolated in this research ranges from 2.19 mg/L to 55.74 mg/L. Compared with the research of Gong Ruihong et al., some IPB strains obtained in this research have better phosphate-solubilizing activity.

In addition, Man Jing et al. [27] isolated and screened 26 strains of IPB from the rhizosphere of *Leymus chinensis* in Lanxi, Xilinhot and Hulun Buir. The strains were inoculated in a medium with $Ca_3(PO_4)_2$ as the sole source of phosphorus and cultured. The concentration of dissolved inorganic phosphorus of these strains was measured to be 7.08–82.71 mg/L. Compared with the research results of Man Jing et al., the phosphate-solubilizing activity of IPB obtained in this research is weaker. Man Jing et al. studied IPB in soil patches containing the rhizosphere of *Leymus chinensis*. In this research, IPB in the rhizosphere of *Hydrilla verticillata* in high-organic matter urban landscape water is studied, and potential new species are found, which enrich the source of phosphate-solubilizing microorganisms. In the future, these IPB strains may be more adaptable to the high-organic matter environment and play a greater role in the treatment of high-organic matter polluted water.

Efficient phosphate-solubilizing strains obtained by laboratory isolation decrease in phosphate-solubilizing efficiency after several generations of transmission [28]. In contrast, in this

research, the single colony is isolated by picking on NBRIP solid medium for delineation and repeatedly delineated for more than four times, and the colony is inoculated in phosphorus-deficient medium for subsequent phosphate-solubilizing characteristic detection experiments after confirmation as pure cultures. The phosphate-solubilizing characteristic of the screened IPB is stable after repeated transgenerational screening.

In this research, we conduct cultivable experiments of IPB in the rhizosphere of *Hydrilla verticillata*, which enriches microbial resources and also determines the phosphate-solubilizing ability of pure cultures, but do not fully understand the full information of IPB species and potential phosphate-solubilizing functions in the roots and rhizosphere sediments of *Hydrilla verticillata*. In the future, experiments on phosphate-solubilizing functional genes, such as GDH, GADH and PQQ [29–31], can be carried out by high-throughput sequencing, which can reflect the distribution of phosphate-solubilizing microorganisms in the environment more comprehensively. Meanwhile, in this research, IPB strains are isolated in the laboratory using NBRIP medium and their phosphate-solubilizing ability is determined using ammonium molybdate spectrophotometry, but no experiments are designed to verify their phosphate-solubilizing ability on sediment and their probiotic effect on the growth of *Hydrilla verticillata*. In the future, it is suggested to select IPB strains with better phosphate-solubilizing effect, and use microcosm experiments to determine their positive effect on the growth of *Hydrilla verticillata*. The experiments can be set up as untreated and treated *Hydrilla verticillata* with IPB, and after a period of incubation, the phosphorus content in different parts of *Hydrilla verticillata* and rhizosphere sediments can be measured to verify whether the IPB could promote the growth of *Hydrilla verticillata* and the release of phosphorus from rhizosphere sediments. The mechanism of the effect of IPB on plant growth also needs to be further investigated. In addition, IPB with the highest phosphate-solubilizing ability in this research can be studied in more detail in the future in order to apply it to the restoration of submerged plants in high-organic matter water.

## Conclusions

A total of 28 cultivable IPB strains are isolated from the rhizosphere of *Hydrilla verticillata* collected from the high–organic matter urban landscape water. These strains are found to belong to two phyla (*Proteobacteria* and *Firmicutes*) and nine genera (*Bacillus*, *Asticcacaulis*, *Acinetobacter*, *Aeromonas*, *Serratia*, *Exiguobacterium*, *Comamonas*, *Ciceribacter* and *Priestia*). Strains SWIH-7, SWIP-6, SWIP-7, SWIP-13, SWIP-15 and SWIP-16 are the potential new species. The results show that the cultivable IPB strains in the rhizosphere of *Hydrilla verticillata* have a rich diversity, expanding the source of phosphate-solubilizing microorganisms.

All the 28 IPB strains could dissolve inorganic phosphorus and have stable phosphate-solubilizing characteristic, but the IPB strains show significantly different phosphate-solubilizing ability. The strain SWIH-9 has the highest phosphate-solubilizing efficiency of 71% and DIP of 55.74 mg/L, while the other strains have phosphate-solubilizing efficiency of 3% to 62% and DIP of 2.19 to 49.26 mg/L. The phosphate-solubilizing ability of the strain is negatively correlated with the pH of the solution. It is preliminarily determined that the strains dissolve insoluble inorganic phosphorus mainly through acid production, and the specific mechanism needs further research.

The IPB strains isolated and screened in this research are rich in diversity, with stable phosphate-solubilizing characteristic and potential new species. These IPB strains are of great significance in the future treatment of high-organic matter polluted water and the ecological restoration of water.

## Author Contributions

**Conceptualization:** Yong Li.

**Data curation:** Xintao Yu, Sidan Gong, Zhilian Gong.

**Formal analysis:** Sidan Gong, Zhilian Gong.

**Funding acquisition:** Yong Li.

**Investigation:** Huan Liu, Xintao Yu.

**Methodology:** Huan Liu, Xintao Yu.

**Project administration:** Yong Li.

**Supervision:** Yong Li, Zhilian Gong.

**Visualization:** Sidan Gong, Zhilian Gong.

**Writing – original draft:** Yong Li, Huan Liu.

**Writing – review & editing:** Yong Li, Huan Liu.

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
