## [Decision Letter · Decision Letter 0]

21 Nov 2023

PONE-D-23-36571Screening and Biodiversity Analysis of Cultivable Inorganic Phosphate–Solubilizing Bacteria in the Rhizosphere of Hydrilla verticillataPLOS ONE

Dear Dr. Li,

Thank you for submitting your manuscript to PLOS ONE. After careful consideration, we feel that it has merit but does not fully meet PLOS ONE’s publication criteria as it currently stands. Therefore, we invite you to submit a revised version of the manuscript that addresses the points raised during the review process.

We look forward to receiving your revised manuscript.

Kind regards,

Estibaliz Sansinenea

Academic Editor

PLOS ONE

Journal Requirements:

4. Please note that funding information should not appear in any section or other areas of your manuscript. We will only publish funding information present in the Funding Statement section of the online submission form. Please remove any funding-related text from the manuscript.

   "This research was funded by the Key Laboratory Open Project of Southwest Jiaotong University (ZD2022210017). This research was funded by Student Research Training Program(2023145)." 

Additional Editor Comments:

Both reviewers have commented that this MS is very interesting and can be approved after minor revision. Therefore, the authors are invited to do all changes suggested by the reviewers to submit again.

Reviewers' comments:

Reviewer's Responses to Questions

**Comments to the Author**

1. Is the manuscript technically sound, and do the data support the conclusions?

Reviewer #1: Yes

Reviewer #2: Yes

2. Has the statistical analysis been performed appropriately and rigorously? 

Reviewer #1: Yes

Reviewer #2: Yes

3. Have the authors made all data underlying the findings in their manuscript fully available?

Reviewer #1: Yes

Reviewer #2: Yes

4. Is the manuscript presented in an intelligible fashion and written in standard English?

Reviewer #1: Yes

Reviewer #2: Yes

5. Review Comments to the Author

Reviewer #1: The authors examine the inorganic phosphate-solubilizing bacteria in the rhizosphere of Hydrilla verticillata, which may transform insoluble inorganic phosphorus in the environment into soluble phosphorus that H. verticillata can directly absorb and consume. This is an interesting investigation that is well-designed and done, with transparent outcomes.

The paper should be updated according to the suggestions listed below:

-The introduction is a little long, and the article's goal is poorly written.

-In the section on material and methods, why did the authors ignore measuring the pH during the quantification test?

-In the Section 3.2, (In the future, IPB with better phosphate-solubilizing………. the experiments were set up as untreated ) Did you carry out this experiment, or did you suggest perspectives?

- There are various grammatical errors in the paper that should be fixed.

Reviewer #2: In this paper, the authors investigate the Inorganic phosphate-solubilizing bacteria (IPB) in the rhizosphere of Hydrilla verticillata which can convert insoluble inorganic phosphorus in the environment into soluble phosphorus that can be directly absorbed and utilized by H. verticillata. This is an interesting study, and it is well-designed and conducted, and the results are clear. The authors have used basic methods to achieve the objective of the study. However, some important points have to be clarified before action can be taken.

The paper needs to be revised according to the following recommendations:

1. There are some grammatical errors throughout the paper; they need to be corrected.

4. In the introduction, Lines 78 to 83 must be moved to the discussion; the introduction should provide a clear rationale for the research and create a solid foundation for the following sections.

2. In section 2.4, authors should add a part in which they explain how they are calculating Phosphate solubilizing Efficiency.

3. In section 2.4, the quantitative test of phosphate solubilization is always accompanied by a drop in pH, therefore why the authors ignored measuring it during the test?

5. In section 3.2, figure 2 is a repetition of column 3 of Table 2, the authors can remove it since it does not give new results.

6. In section 3.2, from line 295 to line 304, the author gives perspectives but the content of the text is the past ‘’In the future, IPB with better phosphate-solubilizing………. The experiments were set up as untreated and treated H……..’’

7. In general, the section results and discussion must be reorganized to eliminate some repetitions.

6. PLOS authors have the option to publish the peer review history of their article (what does this mean?). If published, this will include your full peer review and any attached files.

Reviewer #1: **Yes: **Fatima Zahra ALIYAT

Reviewer #2: **Yes: **IBIJBIJEN J.

---

## [Author Response · Author response to Decision Letter 0]

20 Dec 2023

Reviewer 1

1. The introduction is a little long, and the article's goal is poorly written.

Reply: Accepted and they have been corrected.

2. In the section on material and methods, why did the authors ignore measuring the pH during the quantification test?

Reply: We measured the pH of the culture solution during the experiment, which has been added to the three chapters of Materials and methods, Results and discussion and Conclusions.

3. In the Section 3.2, (In the future, IPB with better phosphate-solubilizing………. the experiments were set up as untreated ) Did you carry out this experiment, or did you suggest perspectives?

Reply: These are our perspectives. They have been corrected in the manuscript.

4. There are various grammatical errors in the paper that should be fixed.

Reply: Accepted and they have been corrected.

Reviewer 2

1. There are some grammatical errors throughout the paper; they need to be corrected.

Reply: Accepted and they have been corrected.

2. In section 2.4, authors should add a part in which they explain how they are calculating Phosphate solubilizing Efficiency.

Reply: Accepted and it has been added. 

3. In section 2.4, the quantitative test of phosphate solubilization is always accompanied by a drop in pH, therefore why the authors ignored measuring it during the test?

Reply: We measured the pH of the culture solution during the experiment, which has been added to the three chapters of Materials and methods, Results and discussion and Conclusions.

4. In the introduction, Lines 78 to 83 must be moved to the discussion; the introduction should provide a clear rationale for the research and create a solid foundation for the following sections.

Reply: Accepted and they have been corrected.

5. In section 3.2, figure 2 is a repetition of column 3 of Table 2, the authors can remove it since it does not give new results.

Reply: Accepted and it has been deleted.

6. In section 3.2, from line 295 to line 304, the author gives perspectives but the content of the text is the past “In the future, IPB with better phosphate-solubilizing……. The experiments were set up as untreated and treated H……”

Reply: Accepted and they have been corrected.

7. In general, the section results and discussion must be reorganized to eliminate some repetitions.

Reply: Accepted and they have been corrected.

---

## [Editor Report · Decision Letter 1]

28 Dec 2023

Screening and biodiversity analysis of cultivable inorganic phosphate–solubilizing bacteria in the rhizosphere of Hydrilla verticillata

PONE-D-23-36571R1

Dear Dr. Li,

We’re pleased to inform you that your manuscript has been judged scientifically suitable for publication and will be formally accepted for publication once it meets all outstanding technical requirements.

Kind regards,

Estibaliz Sansinenea

Academic Editor

PLOS ONE

Additional Editor Comments (optional):

The authors have done all revisions suggested by the reviewers theregore the MS can be accepted
---

## [Editor Report · Acceptance letter]

9 Jan 2024

PONE-D-23-36571R1 

PLOS ONE

Dear Dr. Li, 

I'm pleased to inform you that your manuscript has been deemed suitable for publication in PLOS ONE. Congratulations! Your manuscript is now being handed over to our production team.

Kind regards, 

on behalf of

Dr. Estibaliz Sansinenea 

Academic Editor

PLOS ONE